# Gamma-Poisson Dynamic Matrix Factorization Embedded with Metadata Influence

**Trong Dinh Thac Do**
Advanced Analytics Institute
University of Technology Sydney
thacdtd@gmail.com

**Longbing Cao** *
Advanced Analytics Institute
University of Technology Sydney
longbing.cao@gmail.com

## Abstract

A conjugate Gamma-Poisson model for Dynamic Matrix Factorization incorporated with *m*etadata influence (mGDMF for short) is proposed to effectively and efficiently model *massive*, *sparse* and *dynamic* data in recommendations. Modeling recommendation problems with a massive number of ratings and very sparse or even no ratings on some users/items in a dynamic setting is very demanding and poses critical challenges to well-studied matrix factorization models due to the large-scale, sparse and dynamic nature of the data. Our proposed mGDMF tackles these challenges by introducing three strategies: (1) constructing a stable Gamma-Markov chain model that smoothly drifts over time by combining both static and dynamic latent features of data; (2) incorporating the user/item metadata into the model to tackle sparse ratings; and (3) undertaking stochastic variational inference to efficiently handle massive data. mGDMF is conjugate, dynamic and scalable. Experiments show that mGDMF significantly (both effectively and efficiently) outperforms the state-of-the-art static and dynamic models on large, sparse and dynamic data.

## 1 Introduction

An increasing amount of research [8, 21, 13, 25, 3, 16, 34] focuses on the significant real-life recommendation challenges of modeling massive and evolving ratings (e.g., a girl likes cartoon movies in her childhood but this may change to romantic movies when she is older) but some users (or items) have only a few or or even no ratings (forming sparse or cold-start user/item ratings). For example, Netflix data have $97.5M$ ratings, $225K$ users, $14K$ movies, and 98.8% missing ratings. The intensively-studied collaborative filtering models, in particular matrix factorization (MF) models, fail to model such massive, dynamic and sparse recommendation problems as they usually model static data, assume certain user/item rating similarity, and are too costly to estimate missing ratings in massive data.

Several Poisson-based MF models were proposed recently to model large and sparse static ratings, e.g., Poisson Factorization (PF) [25], and collaborative topic PF for modeling content-based recommendations [27]; and dynamic data, e.g., dynamic PF (dPF) [16], dynamic compound PF (DCPF) [34], and deep dynamic PF [24]. However, none of these can effectively and efficiently handle massive, dynamic and sparse ratings simultaneously (see more analysis in Section 2).

To effectively model sparse, dynamic and massive ratings, we propose a Gamma-Poisson Dynamic Matrix Factorization model incorporated with *m*etadata influence (mGDMF). mGDMF has three built-in mechanisms to jointly address user/item rating sparsity, large-scale ratings, and rating dynamics. First, mGDMF is a factorization model that uses a Gamma-Poisson structure to model massive,

---

sparse and long-tailed data [32, 45, 43]; the Gamma-Poisson structure with Poisson likelihood and non-negative representations enjoys more efficient inference and better handling of sparse data than the Gaussian Factorization in PMF [25, 19, 20].

Second, mGDMF has a conjugate Gamma-Gamma of integrating the observable user/item metadata (e.g., 'age' of a user and 'genre' of a movie) with latent user preference and latent item attractiveness factorized from ratings to model user/item rating sparsity. This is inspired by the observation that rating behaviors are driven by user/item metadata, and the couplings between users/items (i.e., users (or items) with similar metadata may share similar items (or users)) [10, 9, 11]. This metadata-based representation leverages the rating factorization to handle sparse item/user ratings or cold-start rating issues.

Lastly, mGDMF has the conjugate Gamma-Markov chains to model user preferences and item features that change smoothly over time. As a result of jointly handling all three challenges: scalability, sparsity and dynamics, mGDMF forms a conjugate Gamma-Gamma-Gamma-Poisson structure, on which we perform the stochastic variational inference to model massive data.

Extensive empirical results show that mGDMF effectively and efficiently outperforms the state-of-the-art static and dynamic PF models on five large and sparse datasets.

## 2 Related Work

Here, we review the work related to ours, including MF-based models and statistical models for dynamic data and for handling user/item rating sparsity.

**MF-based models.** Classic MF models are improper for handling a large number of ratings [39, 25] as they require intensive mathematical computation and may fail to find similar users in sparse data (they assume two users have rated at least some items in common). Many probabilistic MF models, such as PMF [39], have been proposed to deal with large data. However, data with sparse ratings significantly challenges them since they have to compute all data with many missing ratings.

**Modeling dynamic data.** Several previous studies tend to capture the evolving characteristics of users and items over time, such as TimeSVD++ [36] and Bayesian Probabilistic Tensor (BPTF) Factorization [44]. TimeSVD++ only captures the user-evolving factors but ignores the item-evolving factors. BPTF models the user and item factors at each time index independently from previous ones, and cannot handle specific users/items. The work in [17, 18, 41] extends PMF to dynamic data, but takes the Gaussian state space and cannot handle sparsity as in the long-tail Gamma priors taken in PF [25]. Further, computing on all data (including missing and non-missing elements) makes these models inefficient on large data. Poisson-based dynamic matrix factorization models are recent advances for modeling dynamic data, such as dPF [16] and DCPF [34] for recommendations. dPF faces the same problem as dynamic PMF since it uses the Gaussian state space. DCPF uses the conjugate Gamma-Markov chains but assumes the static portions as a prior of dynamic portions. This makes the chains grow too fast or too slow [34], resulting in unpredictable results. In addition, recent dynamic Poisson-based models such as [24, 47, 40, 1] analyze sequential count vectors. In contrast, mGDMF has the conjugate Gamma-Markov chains and aggregates the static portions with the dynamic portions at each time slice and prevents the instability of the chains. They capture stably evolving user preferences and item attractiveness over time. As a result, it is more efficient and effective to model the nature of dynamic observations.

**Handling user/item rating sparsity.** No work reported directly incorporates user/item metadata into PF for dynamic data. The work in [2, 27, 46, 31] integrates a document-word matrix into PF. Other recent work [48, 22] also tends to integrate observable attributes into some probabilistic models for link prediction, but only works on small data, so sparsity is not addressed there. In addition, SPF [15] and RPF [30] only incorporate the binary relations (0 and 1) of users, however, our method can weight the relations of both users and items. The Gamma-Poisson models in [19, 20] incorporate general attributes for modeling large and sparse data but cannot model dynamic data. mGDMF is the only PF model that incorporates user/item metadata, embedded with more general attributes (e.g., categorical attributes) to work with dynamic data. First, the user/item metadata is modeled as Gamma priors of the latent user preference/latent item attractiveness in the static portion (see 1.(a).ii/1.(b).ii in the mGDMF generative process). Second, these latent user/item features are further given as Gamma priors for the user/item global static factors (as shown in 1.(a).iii/1.(b).iii). Lastly,

we aggregate the above static user/item latent features with the user/item local dynamic factors. This iterative integration process adds more weight to similar items/users, which cannot be captured by any existing PF models.

## 3   The mGDMF Model

For a recommendation problem, we assume the availability of the rating matrix $Y_t$, where each entry is the rating given by user $u$ to item $i$ at time slice $t$ and 0 indicates no rating, and the user metadata $HU$ and item metadata $HI$. The time slice corresponds to the period of time (e.g., months) when users place ratings on items.

Further, we assume the rating matrix $Y_t$ at time slice $t$ follows the Poisson distribution and can be factorized to vectors representing $K$ latent user preferences and vectors representing $K$ latent item attractiveness. The latent user preference vectors and the latent item attractiveness vectors are assumed to combine both static and dynamic portions. The static portions ($\overline{\theta}_{uk}$ for user and $\overline{\beta}_{ik}$ for item) represent the time-independent aspects of users/items, while the dynamic portions ($\theta_{uk,t}$ for user and $\beta_{ik,t}$ for item) capture the time-evolving aspects of users/items.

**Metadata Integration.** The static portions ($\overline{\theta}_{uk}$ and $\overline{\beta}_{ik}$) capture the global stationary (i.e., global static) factors for user $u$ and item $i$, which are not influenced by time, and are assumed to follow the Gamma distribution. We further assume the Gamma distribution of each user's latent preferences, $\xi_u$, and each item's latent attractiveness, $\eta_i$. The influence of user (item) metadata is captured by giving the second parameter (i.e., *rate*) of Gamma distribution of a user's latent preference (item's latent attractiveness) the product of the appearance of the user (item) attribute value in the metadata. The weight (i.e., the importance) of each user attribute, $hu_m$, is given a Gamma prior. The weight of user attribute $hu_m$ affects the preference of a user $\xi_u$ and further affects the static portion of user representations, $\overline{\theta}_{uk}$, if and only if $fu_{u,m} = 1$. We note that $fu_{u,m}$ is binary value that indicates whether user $u$ has the attribute $m$ (i.e., $fu_{u,m} = 1$) or not (i.e., $fu_{u,m} = 0$). $hu_m$ measures the degree of influence of each user attribute, e.g., a user's 'location' may have less influence than the user's 'age' on movie ratings. The weight of an item attribute $hi_n$ is also assumed with a Gamma distribution. $hi_n$ affects the item's latent attractiveness $\eta_i$ and further affects the static portion of item representations, $\overline{\beta}_{ik}$, when item $i$ has the attribute $n$ (i.e., $fi_{i,n} = 1$).

**Dynamic Modelling.** The dynamic portions ($\theta_{uk,t}$ and $\beta_{ik,t}$) serve as the local non-stationary (i.e., local dynamic) factors to capture the evolution of users and items over time. As shown in [14], it is possible to define a Gamma-Markov chain in a straightforward way by $\theta_{uk,t} \sim Gamma(a_\theta, \theta_{uk,t-1}/a_\theta)$. The full conditional distribution $p(\theta_{uk,t}|\theta_{uk,t-1}, \theta_{uk,t+1})$ is conjugate. However, it is not possible to attain a positive correlation between $\theta_{uk,t}$ and $\theta_{uk,t-1}$ since $E[\theta_{uk,t}] = 1/\theta_{uk,t-1}$. Hence, we build a chain that smoothly evolves over time by adding the auxiliary variables $\lambda_{uk,t}$ between $\theta_{uk,t}$ and $\theta_{uk,t-1}$. The auxiliary variables make $E[\theta_{uk,t}] = \theta_{uk,t-1}$. Hence, $\theta_{uk,t}$ increases/decreases when $\theta_{uk,t-1}$ increases/decreases. Operations similar to the above are also taken on the item's dynamic portions.

The generative process of mGDMF is presented below and the graphical model of mGDMF can be found in the supplementary.

1. **Metadata Integration:**
    (a) For each user:
        i. Draw the weight of $m^{th}$ attribute in user metadata $hu_m \sim Gamma(a', b')$
        ii. Draw latent user preference $\xi_u \sim Gamma(a, \prod_{m=1}^{M} hu_m^{fu_{u,m}})$
        iii. Draw global static factor $\overline{\theta}_{uk} \sim Gamma(b, \xi_u)$
    (b) For each item:
        i. Draw the weight of $n^{th}$ attribute in item metadata $hi_n \sim Gamma(c', d')$
        ii. Draw latent item attractiveness $\eta_i \sim Gamma(c, \prod_{n=1}^{N} hi_n^{fi_{i,n}})$
        iii. Draw global static factor $\overline{\beta}_{ik} \sim Gamma(d, \eta_i)$
2. **Dynamic Modeling:**
    (a) For each user:

    i. Draw initialized state of local dynamic factor $\theta_{uk,1} \sim Gamma(a_\theta, a_\theta b_\theta)$

    ii. For each time slice $t > 1$:

        A. Draw auxiliary variable $\lambda_{uk,t-1} \sim Gamma(a_\lambda, a_\lambda \theta_{uk,t-1})$

        B. Draw local dynamic factor $\theta_{uk,t} \sim Gamma(a_\theta, a_\theta \lambda_{uk,t-1})$

(b) For each item:

    i. Draw initialized state of local dynamic factor $\beta_{ik,1} \sim Gamma(a_\beta, a_\beta b_\beta)$

    ii. For each time slice $t > 1$:

        A. Draw auxiliary variable $\iota_{ik,t-1} \sim Gamma(a_\iota, a_\iota \beta_{ik,t-1})$

        B. Draw local dynamic factor $\beta_{ik,t} \sim Gamma(a_\beta, a_\beta \iota_{ik,t-1})$

3. **For each rating:**

(a) Draw $y_{ui,t} \sim Poisson(\sum_k (\theta_{uk,t} + \overline{\theta}_{uk})(\beta_{ik,t} + \overline{\beta}_{ik}))$

As a result, mGDMF effectively models both static and dynamic characteristics of user preference and item attractiveness in the context of having sparse ratings on users/items.

**Handling massive data.** We further describe how mGDMF models massive data. We calculate the probability of rating $y_{ui,t}$ by user $u$ on item $i$ at time slice $t$ as:

$$p(y_{ui}|\overline{\theta}_u, \theta_u, \overline{\beta}_i, \beta_i) = \frac{((\overline{\theta}_u + \theta_u)(\overline{\beta}_i + \beta_i))^{y_{ui}} exp\{-((\overline{\theta}_u + \theta_u)(\overline{\beta}_i + \beta_i)\}}{y_{ui}!} \qquad (1)$$

When $y_{ui} = 0$, it does not affect the probability. Owing to the Poisson factorization [25], it does not require optimization to reduce the computational time as in the classical MF [38]. The probability only depends on $\overline{\theta}_u, \theta_u, \overline{\beta}_i$ and $\beta_i$.

**Better prediction with metadata integration.** Richer priors are provided by integrating user metadata to represent the user's latent preference $\xi_u$ as in Eq. (2). This enhanced user's latent preference representation $\xi_u$ in turn provides richer priors to the user's global static factor $\overline{\theta}_{uk}$.

$$\xi_u|\overline{\theta} \sim Gamma(a + Kb, \prod_{m=1}^{M} hu_m^{fu_{u,m}} + \sum_k \overline{\theta}_{uk}) \qquad (2)$$

The global static factors then affect the time-sensitive local dynamic factors. Similarly, we integrate item metadata into representing latent item attractiveness and its evolution. As a result, mGDMF integrates both observable user/item metadata and latent static/dynamic portions.

## 4 Stochastic Variational Inference for mGDMF

We first apply the mean-field Variational Inference (VI) [42] to the approximate inference of the posterior distribution, which is shown [42] to be more efficient than methods like MCMC [23] for large-scale probabilistic models. The mean-field VI chooses a family of variational distributions over all hidden variables. The posteriors of all variational distributions are then approximated by tuning the parameters to minimize the Kullback-Leibler divergence to the true posterior.

Given the rating tables $Y_t$ and the user/item metadata $HU$ and $HI$, we compute the posterior distributions of the weight of each user attribute in metadata $hu_m$, the weight of each item attribute in metadata $hi_n$, the user's latent preference $\xi_u$ (expressed as the static global factor $\overline{\theta}_{uk}$ and local dynamic factor $\theta_{uk,t}$), and the item's latent attractiveness $\eta_i$ (represented as the static global factor $\overline{\beta}_{uk}$ and local dynamic factor $\beta_{ik,t}$).

To ensure the conjugacy of the model structure, inspired by [25, 21, 49, 26], the rating $y_{ui,t}$ is replaced with auxiliary latent variable $z_{ui,k,t} \sim Poisson((\theta_{uk,t} + \overline{\theta}_{uk})(\beta_{ik,t} + \overline{\beta}_{ik}))$. With the additive property of Poisson distribution, $y_{ui,t}$ is expressed as $y_{ui,t} = \sum_k z_{ui,k,t}$. Then, the mean-field family assumes each distribution is independent of each other and is governed by its own

distribution.

$$q(hu, hi, \xi, \eta, \overline{\theta}, \overline{\beta}, \lambda, \iota, \theta, \beta, z) = \prod_m q(hu_m|\zeta_m) \prod_n q(hi_n|\rho_n) \prod_u q(\xi_u|\kappa_u) \prod_i q(\eta_i|\tau_i)$$

$$\prod_{u,k} q(\overline{\theta}_{uk}|\overline{\nu}_{uk}) \prod_{i,k} q(\overline{\beta}_{ik}|\overline{\mu}_{ik}) \prod_{u,k,t} q(\theta_{uk,t}|\nu_{uk,t}) \prod_{i,k,t} q(\beta_{ik,t}|\mu_{ik,t}) \qquad (3)$$

$$\prod_{u,k,t} q(\lambda_{uk,t}|\gamma_{uk,t}) \prod_{i,k,t} q(\iota_{ik,t}|\omega_{ik,t}) \prod_{u,i,t,k} q(z_{ui,t,k}|\phi_{ui,t,k})$$

We use the class of conditionally conjugate priors for $hu_m$, $hi_n$, $\xi_u$, $\eta_i$, $\overline{\theta}_{uk}$, $\overline{\beta}_{ik}$, $\theta_{uk}$, $\lambda_{uk,t}$, $\beta_{ik}$, $\iota_{ik,t}$ and $z_{ui,t,k}$ to update the variational parameters $\{\zeta, \rho, \kappa, \tau, \overline{\nu}, \overline{\mu}, \nu, \gamma, \mu, \omega, \phi\}$. For the Gamma distribution, we update both hyper-parameters: *shape* and *rate*.

Table 1: Latent Variables, Type, Variational Variables and Variational Update for Users. Similar variables for items (i.e., $hi_n$, $\eta_i$, $\overline{\beta}_{ik}$, $\beta_{ik}$, $\iota_{ik,t}$) can be found in the supplementary. $\aleph_m$ is the number of users having the $m^{th}$ attribute, $K$ is the number of latent components, and $\Psi(.)$ is the *digamma* function. The Gamma distribution is parameterized by *shape* ($shp$) and *rate* ($rte$).

| Latent Variable | Type | Variational Variable | Variational Update |
|---|---|---|---|
| $hu_m$ | Gamma | $\zeta_m^{shp}, \zeta_m^{rte}$ | $a' + \aleph_m a,\ b' + \sum_u \frac{\kappa_u^{shp}}{\kappa_u^{rte}}$ |
| $\xi_u$ | Gamma | $\kappa_u^{shp}, \kappa_u^{rte}$ | $a + Kb,\ \prod_{m=1}^M \left(\frac{\zeta_m^{shp}}{\zeta_m^{rte}}\right)^{fu_{u,m}} + \sum_k \frac{\overline{\nu}_{uk}^{shp}}{\overline{\nu}_{uk}^{rte}}$ |
| $z_{ui,t,k}$ | Mult | $\phi_{ui,t,k}$ | $(exp\{\Psi(\nu_{uk,t}^{shp}) - log(\nu_{uk,t}^{rte})\} + exp\{\Psi(\overline{\nu}_{uk}^{shp}) - log(\overline{\nu}_{uk}^{rte})\})$ $* (exp\{\Psi(\mu_{ik,t}^{shp}) - log(\mu_{ik,t}^{rte}) + exp\{\Psi(\overline{\mu}_{ik}^{shp}) - log(\overline{\mu}_{ik}^{rte}))\})$ |
| $\overline{\theta}_{uk}$ | Gamma | $\overline{\nu}_{uk}^{shp}, \overline{\nu}_{uk}^{rte}$ | $b + \sum_{i,t} y_{ui,t}\phi_{ui,t,k},\ \frac{\kappa_u^{shp}}{\kappa_u^{rte}} + \sum_i \left(\frac{\overline{\mu}_{ik}^{shp}}{\overline{\mu}_{ik}^{rte}} + \sum_t \frac{\mu_{ik,t}^{shp}}{\mu_{ik,t}^{rte}}\right)$ |
| $\theta_{uk,t}$ | Gamma | $\nu_{uk,t}^{shp}$ | $a_\theta + a_\lambda + \sum_i y_{ui,t}\phi_{ui,t,k}$ |
| | | $\nu_{uk,1}^{rte}$ | $a_\theta b_\theta + a_\lambda \frac{\gamma_{uk,1}^{shp}}{\gamma_{uk,1}^{rte}} + \sum_i \left(\frac{\overline{\mu}_{ik}^{shp}}{\overline{\mu}_{ik}^{rte}} + \frac{\mu_{ik,1}^{shp}}{\mu_{ik,1}^{rte}}\right)$ |
| | | $\nu_{uk,t,(t>1)}^{rte}$ | $a_\theta \frac{\gamma_{uk,t-1}^{shp}}{\gamma_{uk,t-1}^{rte}} + a_\lambda \frac{\gamma_{uk,t}^{shp}}{\gamma_{uk,t}^{rte}} + \sum_i \left(\frac{\overline{\mu}_{ik}^{shp}}{\overline{\mu}_{ik}^{rte}} + \frac{\mu_{ik,t}^{shp}}{\mu_{ik,t}^{rte}}\right)$ |
| $\lambda_{uk,t}$ | Gamma | $\gamma_{uk,t}^{shp}, \gamma_{uk,t}^{rte}$ | $a_\lambda + a_\theta,\ a_\lambda \frac{\nu_{uk,t}^{shp}}{\nu_{uk,t}^{rte}} + a_\theta \frac{\nu_{uk,t+1}^{shp}}{\nu_{uk,t+1}^{rte}}$ |

With the mean-field VI, the coordinate ascent is used to iteratively optimize each variational parameter while holding the others fixed [35]. The full variational parameter update is shown in Table 1 and the batch algorithm can be found in the supplementary material.

**Stochastic Algorithm.**

Mean-field VI iterates to update variational parameters by involving the entire data at each iteration until convergence to a local optimum, which could be computationally intensive for large data.

We thus adopt stochastic optimization by sampling a data point from the rating $y_{ui,t}$ of user $u$ on item $i$ to update its local parameters as in batch inference, and then update the global variables (similar to [29]). For example, to update the *shape* of a user's dynamic factors, we form the intermediate *shape* of a user's dynamic factors with the sampled rating's optimized local parameters as follows:

$$\nu_{uk,t}^{shp(imd)} = a_\theta + a_\lambda + I.y_{ui,t}\phi_{ui,t,k} \qquad (4)$$

where $I$ is the total number of items in the dataset.

We then update the global variational parameters by taking step $\epsilon$ in the direction of the stochastic natural gradient.

$$\nu_{uk,t}^{shp(iter+1)} = (1 - \sigma_{iter})\nu_{uk,t}^{shp(iter)} + \sigma_{iter}\nu_{uk,t}^{shp(imd)} \qquad (5)$$

where $\sigma_{iter} > 0$ is a step size at iteration $iter$. As shown in [7, 29], to ensure convergence, one possible choice of $\sigma_{iter}$ is $(iter_0 + iter)^{-\epsilon}$ for $iter_0 > 0$ and $\epsilon \in (0.5, 1]$. $iter_0$ and $\epsilon$ are called the *learning rate delay* and the *learning rate power* respectively.

---

**Algorithm 1** SVI for mGDMF

---

Initialize $\{\zeta, \rho, \kappa, \tau, \overline{\nu}, \overline{\mu}, \nu, \mu, \gamma, \omega, \phi\}$.
Set $K$: # latent components, $U$: # users, $I$: # items, $iter_0$ and $\epsilon$.
**repeat**
    **for** each time slice $t = 1...T$ **do**
        Sample a rating $y_{ui,t}$ uniformly from the dataset.
        Update the local variational parameter of multivariate parameter $\phi$.
        Update all intermediate variational parameters similar to Eq. (4).
        Update all global variational parameters similar to Eq. (5).
        Update the learning rates $iter$.
    **end for**
**until** convergence

---

The updating of the other global variational parameters is similar to that of the user's dynamic factors. The SVI algorithm for mGDMF is presented in Algorithm 1.

**Computational efficiency.** The SVI is more efficient than batch inference algorithms in terms of working with massive data. The computational cost per iteration of batch is $O(U_0 + I_0)K$, where $U_0$ and $I_0$ are the non-zero observations in the rating matrix. It is more efficient than classic PMF with the computational cost $O(U + I)K$ ($U$ and $I$ are the total numbers of user/item observations respectively). The computational cost of the SVI algorithm per iteration is $O(U_s + I_s)K$, $U_s$ and $I_s$ are the non-zero observations sampled from users and items respectively.

## 5 Experiments

Here, mGDMF is evaluated in terms of its capability to model dynamic data and data with sparse user/item ratings and its efficiency in handling large data.

### 5.1 Datasets

GDMF/mGDMF are tested on the following five public dynamic datasets with massive and sparse/cold-start ratings and some metadata.

**Netflix-Time.** Similar procedure as in [37, 16, 34] is taken to obtain a subset of Netflix Prize data [4] with only active users and movies between 1998 and 2005. It contains $10.4K$ users, $6.5K$ movies and $2.5M$ non-missing ratings (from 1 to 5) over 16 time slices (with the duration of each time slice as 3 months) with the metadata: the 'year of release' of the movies.

**Netflix-Full.** We also fit our models on the whole Netflix dataset: $225K$ users, $14K$ movies and $6.9M$ observations over 14 time slices (with the duration of each time slice as 6 months) with the metadata: the 'year of release' of the movies. This data challenges inference and analysis since its ratings distribution is extremely long-tailed and the users are very inactive. It tests the effect of modeling sparse items/users when a large number of items are associated with only few (or no) ratings from users.

**Yelp-Active.** A subset of the Yelp Academic Challenge data is obtained similarly to [34]: $0.9K$ active customers and $6.7K$ business units between 2008 and 2015 over 12 time slices (with the duration of each time slice as 6 months). It includes user metadata 'year join' and 'average star', and the item metadata 'location', 'star', 'take out' (0 or 1), and 'parking' (0 or 1).

**LFM-Tracks.** It contains the number of times a user listened to a song during a given time period [12]: 16 time slices of $0.9K$ users and $1M$ tracks (i.e., songs), similar to [34]; user metadata includes 'age' (partitioned to ranges: $1 :$ "$Under\ 18$", $18 : $ "$18 - 24$", $25 : $ "$25 - 34$", $35 : $ "$35 - 44$", $45 :$ "$45 - 49$", $50 : $ "$50 - 55$", $and\ 56 : $ "$56 +$"), 'gender', 'country' and 'sign up year'; and item metadata includes 'genre' of the music derived from the 'tag' of each track (e.g., "rock", "pop" or "alternative").

**LFM-Bands.** We consider bands instead of tracks as items in LFM-Tracks: $0.9K$ users and $107K$ bands with the same metadata as LFM-Tracks.

## 5.2 Baseline Methods

We compare mGDMF with both state-of-the-art static and dynamic PF models in handling dynamic data. The mGDMF without metadata integration is a Gamma-Poisson Dynamic Matrix Factorization (GDMF) model to test the effect of metadata integration. GDMF replaces the *rate* of Gamma distribution of latent user preference $\xi_u$ by hyper-parameter $a$ and the item's latent attractiveness $\eta_i$ by hyper-parameter $c$ instead of the product of the user/item attribute's weights.

**Static Models.** As shown in [25], HPF outperforms baselines including Non-negative Matrix Factorization [5], LDA [6], PMF [39], and basic PF [8, 21, 13], where HCPF [3] is the latest static model in the PF family. We thus compare mGDMF with two different versions of HPF and HCPF to demonstrate the mGDMF benefits in modeling the time-evolving effect. Similar to [16], HPF-all and HCPF-all are trained on all training ratings while HPF-last and HCPF-last are trained by using the last time slice in the training set as the observations.

**Dynamic Models.** The two most recently reported PF models for recommendation are dPF [16] and DCPF [34]. dPF was shown to outperform state-of-the-art dynamic collaborative filtering algorithms, specifically, BPTF [44] and TimeSVD++ [36]. Therefore, we show the advantage of mGDMF with the metadata integration and Gamma-Poisson structure by comparing it with dPF and DCPF.

## 5.3 Experiment Settings and Evaluation

**Initialization.** For the static portions, we set $a = b = c = d = 0.3$ in the same way as in HPF. The metadata hyper-parameters $a'$, $b'$, $c'$ and $d'$ are set to a small value: 0.1, so that the user/item attribute weights automatically grow over time. We also set $a_\theta = a_\gamma = a_\theta = b_\theta = b_\beta = a_\iota = 1$ to keep the chains small at the beginning. We test a wide range of latent components $K$ from 10 to 200 and choose the best $K = 50$ for mGDMF/GDMF. For SVI hyper-parameters, we assign $10,000$ as the learning rate delay $iter_0$ and $0.7$ as the learning rate power $\epsilon$, similar to [34] and [3].

**Evaluation Metrics.** We hold out the last time slice for testing and keep the previous slices for training, i.e., at each slice $t$, the observations are the data before $t$ (i.e., $1 \ldots t-1$) and the test set includes the ratings at $t$. The results are the average of all slices $t$ from 2 to the maximal slice $T$. We then randomly sample and assign $5\%$ of the test set for validation, similar to [16, 34]. The top-$N$ recommendations are obtained in training w.r.t. the highest prediction score. In testing, we compute the *precision*@$N$, which measures the fraction of relevant items in a user's top-$N$ recommendations, and *recall*@$N$, which is the fraction of the testing items that present in the top-$N$ recommendations. We also calculate the same evaluation metrics as DCPF: Normalized Discounted Cumulative Gain (NDCG) [33] and the Area Under the ROC Curve (AUC) [28].

**Prediction.** We test a practical scenario with given past ratings to predict the ratings at the current time slice $t$, which is ranked by their posterior-expected Poisson parameters as follows.

$$score_{ui,t} = \sum_k E_{posterior}[(\theta_{uk,t} + \overline{\theta}_{uk})(\beta_{ik,t} + \overline{\beta}_{ik})] \tag{6}$$

**Convergence.** Convergence is measured by computing the prediction accuracy on the validation set. mGDMF stops when the change of prediction accuracy w.r.t. log likelihood is less than 0.0001%.

## 5.4 Results

**Effect of metadata and dynamic data modeling.**

Figure 1 reports the results w.r.t. *Precision*@$50$ and *Recall*@$50$ of mGDMF together with GDMF without metadata integration, compared to DCPF, dPF and four static models: HCPF-all, HCPF-last, HPF-all and HPF-last. It is clear that dynamic models (mGDMF, GDMF, DCPF and dPF) are more effective than the static ones (HCPF-all, HCPF-last, HPF-all and HPF-last) on all datasets. Our models both with or without metadata perform the best of the dynamic models. mGDMF gains as much as $10\%$ improvement on Netflix-Full which has an extremely long-tailed rating distribution (i.e., many sparse items/users). mGDMF and GDMF make a large performance difference on Yelp-Active which has the richest metadata. GDMF efficiently models dynamic data by gaining an average $5\%$ over DCPF and $4.6\%$ over dPF w.r.t. *Precision*@$50$ on five datasets. mGDMF effectively integrates metadata by further gaining an average $1.2\%$ on top of our GDMF. The results are consistent with this w.r.t. NDCG/AUC as shown in Table 2, where mGDMF gains maximally 240.69% and minimally

38.38% NDCG improvement and maximally 27.12% and minimally 2.76% AUC improvement over baselines on Yelp-Active.

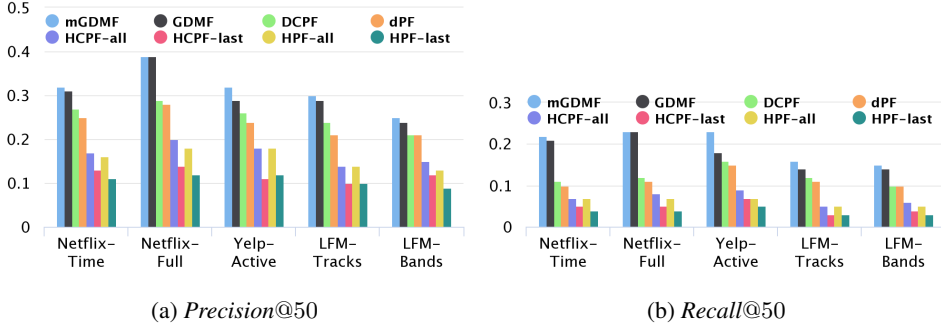

(a) *Precision*@50            (b) *Recall*@50

Figure 1: Top-50 Recommendations Compared with Baselines.

Table 2: Predictive Performance on Five Datasets w.r.t. NDCG and AUC.

| | Netflix-Time | | Netflix-Full | | Yelp-Active | | LFM-Tracks | | LFM-Bands | |
|---|---|---|---|---|---|---|---|---|---|---|
| | NDCG | AUC | NDCG | AUC | NDCG | AUC | NDCG | AUC | NDCG | AUC |
| **mGDMF** | **0.389** | **0.9145** | **0.403** | **0.9321** | **0.494** | **0.8650** | **0.310** | **0.8245** | **0.367** | **0.8217** |
| **GDMF** | 0.367 | 0.9121 | 0.398 | 0.9320 | 0.416 | 0.8512 | 0.275 | 0.8101 | 0.354 | 0.8139 |
| DCPF | 0.293 | 0.9023 | 0.315 | 0.8991 | 0.357 | 0.8418 | 0.231 | 0.8098 | 0.275 | 0.8011 |
| dPF | 0.257 | 0.9012 | 0.301 | 0.8901 | 0.332 | 0.8321 | 0.210 | 0.8019 | 0.298 | 0.8122 |
| HCPF-all | 0.241 | 0.8012 | 0.245 | 0.8370 | 0.243 | 0.8032 | 0.209 | 0.7010 | 0.213 | 0.7121 |
| HCPF-last | 0.183 | 0.7423 | 0.201 | 0.7600 | 0.172 | 0.7312 | 0.132 | 0.5893 | 0.160 | 0.6101 |
| HPF-all | 0.231 | 0.8035 | 0.250 | 0.8124 | 0.248 | 0.8130 | 0.179 | 0.7084 | 0.184 | 0.7013 |
| HPF-last | 0.162 | 0.7213 | 0.198 | 0.7540 | 0.145 | 0.6810 | 0.143 | 0.6050 | 0.141 | 0.5982 |
| $\delta_{min}(\%)$ | 32.76 | 1.35 | 27.94 | 3.67 | 38.38 | 2.76 | 34.20 | 1.82 | 23.15 | 1.70 |
| $\delta_{max}(\%)$ | 140.12 | 26.78 | 103.54 | 23.62 | 240.69 | 27.12 | 134.85 | 44.83 | 160.28 | 37.36 |

**Effect of handling sparse users/items and the 'cold-start' problem.** We report 10 users with the most precisely recommended items and the percentage of precisely recommended sparse items in the top-100 recommendations to compare mGDMF and GDMF with DCPF on LFM-Tracks which has the most dynamic and richest metadata. The sparse item is calculated as the fraction of the non-missing ratings for that in the total number of users (rows). For those items with $< 6\%$ ratings (i.e., sparse items), Figure 2 shows that mGDMF recommends more (on average $7.6\%$ items per user) than DCPF (on average $3.1\%$) of these sparse items to the 10 users. For those items with sparsity $< 1\%$, mGDMF recommends on average $1.6\%$ items per user compared to $0.2\%$ by DCPF, attributed to the richer priors and the mGDMF models with metadata integration and aggregating static and dynamic portions. Further, while DCPF cannot recommend any items when sparsity $= 0\%$ (i.e., cold-start items), mGDMF has recommended on average $0.5\%$ items. In addition, Figure 2 shows that GDMF recommends more sparse items than DCPF and is also more efficient, since the static portions aggregate the dynamic portions. However, GDMF recommends less than our mGDMF especially on items with sparsity $= 0\%$ (i.e., cold-start items). This shows our model's advantage in incorporating metadata.

**Case study of mGDMF-based recommendation.** We illustrate the contributions of mGDMF w.r.t. making recommendations on data Last.fm, shown in Figure 3, where two users '$U270$' and '$U437$' have the same metadata ('age', 'gender', 'country' and 'sign up year'), hence they have similar hobbies as shown on the right of Figure 3. From the dynamic perspective, since two users change their interest from 'pop' to 'rock' music, it is unreasonable to continue to recommend pop music to user $U270$. However, the static models continue to recommend pop music as the number of times the users listened to pop and rock are similar, as shown on the right of Figure 3. mGDMF makes better recommendations (8 out of 10 are in the 'rock' category and no 'pop' track recommended). Further, $U270$ has not listened to 'Zombie' in the past but we can still precisely recommend 'Zombie' because it is in the list of tracks listened to by users with similar metadata to $U437$.

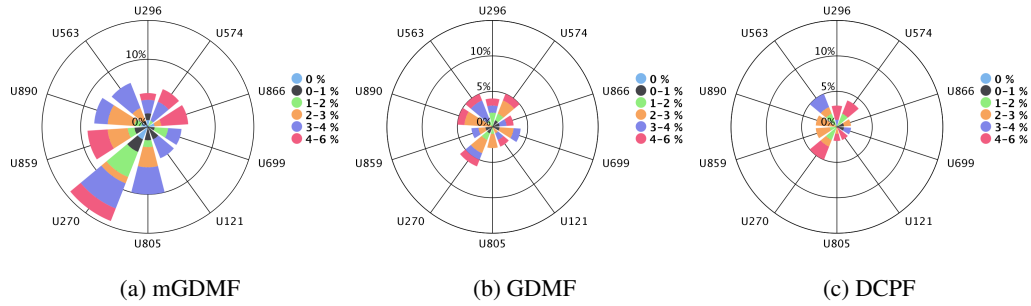

(a) mGDMF        (b) GDMF        (c) DCPF

Figure 2: Percentage (%) of Sparse Items Recommended Precisely for 10 Users by mGDMF, GDMF and DCPF.

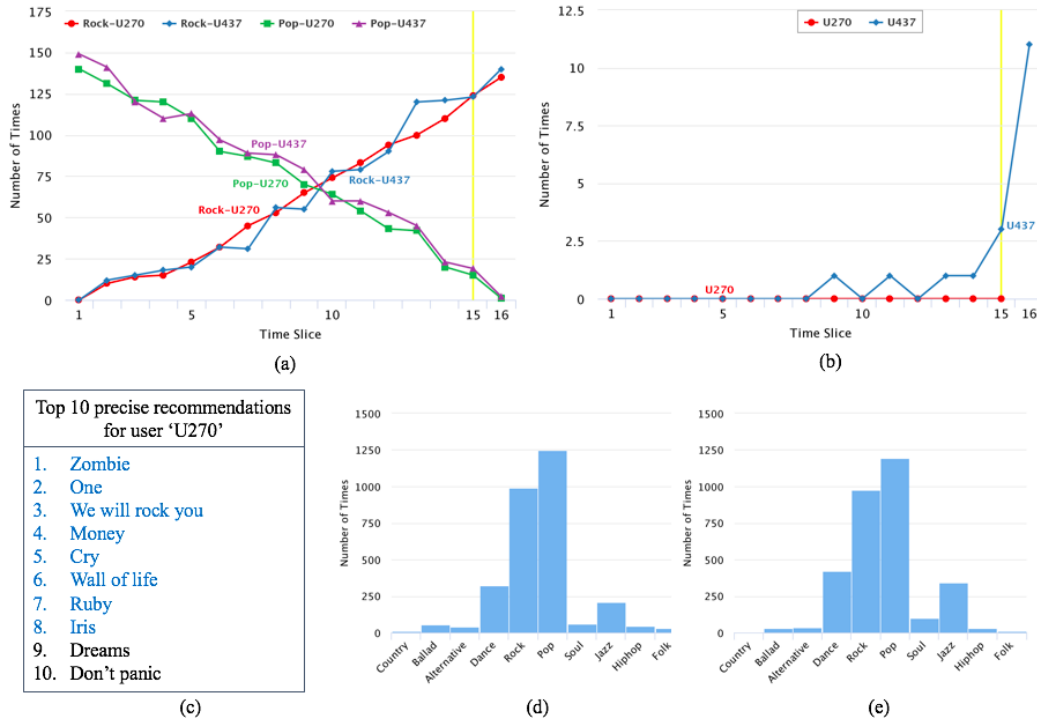

Figure 3: Analysis on two users '$U270$' and '$U437$' with the same metadata in Last.fm. The number of times that users listened to two 'rock' and 'pop' tracks with 16 time slices is shown in (a). The number of times listened to 'Zombie' track by two users '$U270$' and '$U437$' through 16 time slices is shown in (b). The top10 precise recommendations by mGDMF are shown in (c). The distribution of the number of times that $U270$ and $U437$ listened to top 10 'rock' and 'pop' tracks are shown in (d) and (e).

# 6 Conclusions

We proposed a novel dynamic PF model: Gamma-Poisson Dynamic Matrix Factorization with Metadata Influence (mGDMF) to effectively and efficiently model three major challenges in real-life massive recommendations: massive data with sparse and evolving ratings. mGDMF significantly outperforms the state-of-the-art static and dynamic models on five large datasets. mGDMF can further tackle massive, sparse and evolving data by involving time-dependent metadata for scalable recommendation and tackling challenges in other applications such as context-aware chatbot by involving textual and sequential information.

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
