[Supplementary Material]

# Supplementary for Gamma-Poisson Dynamic Matrix Factorization Embedded with Metadata Influence

**Trong Dinh Thac Do**
Advanced Analytics Institute
University of Technology Sydney
thacdtd@gmail.com

**Longbing Cao**
Advanced Analytics Institute
University of Technology Sydney
longbing.cao@gmail.com

Here we present additional details to complement with the paper in terms of the graphical model of mGDMF, the variational update for item's latent variables, the batch inference algorithm, the full derivation of mGDMF.

## 1 Graphical Model of mGDMF

We show the graphical model of mGDMF in Figure 1

Figure 1: Graphical Model of mGDMF

## 2 Variational Update for Items Latent Variables

We show the latent variables for items (i.e., $hi_n$, $\eta_i$, $\overline{\beta}_{ik}$, $\beta_{ik}$, $\iota_{ik,t}$) with their variational update as mentioned in Table 1 in the paper.

Table 1: Latent Variables, Type, Variational Variables and Variational Update for Items (i.e., $hi_n$, $\eta_i$, $\overline{\beta}_{ik}$, $\beta_{ik}$, $\iota_{ik,t}$). $\chi_n$ is the number of items having the $n^{th}$ attribute and $K$ is the number of latent components.

| Latent Variable | Type | Variational Variable | Variational Update |
|:---:|:---:|:---:|:---:|
| $hi_n$ | Gamma | $\rho_n^{shp}$ | $c' + \chi_n c$ |
| | | $\rho_n^{rte}$ | $d' + \sum_i \frac{\tau_i^{shp}}{\tau_i^{rte}}$ |
| $\eta_i$ | Gamma | $\tau_i^{shp}$ | $c + Kd$ |
| | | $\tau_i^{rte}$ | $\prod_{n=1}^{N}(\frac{\rho_n^{shp}}{\rho_n^{rte}})^{fi_{i,n}} + \sum_k \frac{\overline{\mu}_{ik}^{shp}}{\overline{\mu}_{ik}^{rte}}$ |
| $\overline{\beta}_{ik}$ | Gamma | $\overline{\mu}_{ik}^{shp}$ | $d + \sum_{u,t} y_{ui,t}\phi_{ui,t,k}$ |
| | | $\overline{\mu}_{ik}^{rte}$ | $\frac{\tau_i^{shp}}{\tau_i^{rte}} + \sum_u(\frac{\overline{\nu}_{uk}^{shp}}{\overline{\nu}_{uk}^{rte}} + \sum_t \frac{\nu_{uk,t}^{shp}}{\nu_{uk,t}^{rte}})$ |
| $\beta_{ik,t}$ | Gamma | $\mu_{ik,t}^{shp}$ | $a_\beta + a_\iota + \sum_u y_{ui,t}\phi_{ui,t,k}$ |
| | | $\mu_{ik,1}^{rte}$ | $a_\beta b_\beta + a_\iota \frac{\omega_{ik,1}^{shp}}{\omega_{ik,1}^{rte}} + \sum_u(\frac{\overline{\nu}_{uk}^{shp}}{\overline{\nu}_{uk}^{rte}} + \frac{\nu_{uk,1}^{shp}}{\nu_{uk,1}^{rte}})$ |
| | | $\mu_{ik,t,(t>1)}^{rte}$ | $a_\beta \frac{\omega_{ik,t-1}^{shp}}{\omega_{ik,t-1}^{rte}} + a_\iota \frac{\omega_{ik,t}^{shp}}{\omega_{ik,t}^{rte}} + \sum_u(\frac{\overline{\nu}_{uk}^{shp}}{\overline{\nu}_{uk}^{rte}} + \frac{\nu_{uk,t}^{shp}}{\nu_{uk,t}^{rte}})$ |
| $\iota_{ik,t}$ | Gamma | $\omega_{ik,t}^{shp}$ | $a_\iota + a_\beta$ |
| | | $\omega_{ik,t}^{rte}$ | $a_\iota \frac{\mu_{ik,t}^{shp}}{\mu_{ik,t}^{rte}} + a_\beta \frac{\mu_{ik,t+1}^{shp}}{\mu_{ik,t+1}^{rte}}$ |

# 3 Batch Inference Algorithm

The mean-field Variational Inference of mGDMF as mentioned in Section 4 in the paper is shown below.

---

**Algorithm 1** Batch Inference for mGDMF

---

1: Initialize $\{\zeta, \rho, \kappa, \tau, \overline{\nu}, \overline{\mu}, \nu, \mu, \gamma, \omega, \phi\}$.
2: Set $K$: # latent components.
3: Fix the parameters of the *shape* of Gamma distributions of $\zeta, \rho, \gamma, \omega$.
4: **repeat**
5:     **for** each time slice $t = 1...T$ **do**
6:         **for** each rating of user $u$ to item $i$ that $y_{ui} \neq 0$ **do**
7:             Update the multinominal $z_{ui,t}$.
8:         **end for**
9:         **for** each user $u$ **do**
10:            Update the static global factors $\overline{\theta}_{uk}$.
11:            Update *rate* of behavior $\xi_u$.
12:            **for** each user's attribute in metadata $hu_m$ **do**
13:                Update *rate* of the weight.
14:            **end for**
15:            Update the *shape* and *rate* of dynamic correction factors $\theta_{uk,t}$.
16:            Update *shape* of user's auxiliary variables $\lambda_{uk,t}$.
17:         **end for**
18:         **for** each item $i$ **do**
19:            Update the global factors $\overline{\beta}_{ik}$.
20:            Update *rate* of latent attractiveness $\eta_i$.
21:            **for** each item's attribute in metadata $hi_n$ **do**
22:                Update *rate* of the weight.
23:            **end for**
24:            Update the *shape* and *rate* of dynamic correction factors $\beta_{ik,t}$.
25:            Update *shape* of item's auxiliary variables $\lambda_{uk,t}$.
26:         **end for**
27:     **end for**
28: **until** convergence

---

# 4 The Derivation of mGDMF

The mean-field family assumes each distribution is independent of each other and is governed by its own distribution.

$$q(hu, hi, \xi, \eta, \overline{\theta}, \overline{\beta}, \lambda, \iota, \theta, \beta, z) = \prod_m q(hu_m|\zeta_m) \prod_n q(hi_n|\rho_n) \prod_u q(\xi_u|\kappa_u) \prod_i q(\eta_i|\tau_i)$$

$$\prod_{u,k} q(\overline{\theta}_{uk}|\overline{\nu}_{uk}) \prod_{i,k} q(\overline{\beta}_{ik}|\overline{\mu}_{ik}) \prod_{u,k,t} q(\theta_{uk,t}|\nu_{uk,t}) \prod_{i,k,t} q(\beta_{ik,t}|\mu_{ik,t}) \quad (1)$$

$$\prod_{u,k,t} q(\lambda_{uk,t}|\gamma_{uk,t}) \prod_{i,k,t} q(\iota_{ik,t}|\omega_{ik,t}) \prod_{u,i,t,k} q(z_{ui,t,k}|\phi_{ui,t,k})$$

## 4.1 Derivation of Metadata Integration

The derivation of the metadata integration is given here. With the Gamma distribution of user/item's metadata:

$$p(hu_m|a', b') \propto hu_m^{a'-1} exp\{-b' * hu_m\} \quad (2)$$

$$p(hi_n|c', d') \propto hi_n^{c'-1} exp\{-d' * hi_n\} \quad (3)$$

With the Gamma distribution of latent user preference and latent item attractiveness:

$$p(\xi_u|a, hu_m) \propto \left( \prod_{m=1}^{M} hu_m^{fu_{u,m}a} \right) exp\{-(\prod_{m=1}^{M} hu_m^{fu_{u,m}})\xi_u\} \tag{4}$$

$$p(\eta_i|c, hi_n) \propto \left( \prod_{n=1}^{N} hi_n^{fi_{i,n}c} \right) exp\{-(\prod_{n=1}^{N} hi_n^{fi_{i,n}})\eta_i\} \tag{5}$$

The posterior probability of weight $hu_m$ and $hi_n$ become:

$$p(hu_m|a', b', \xi_u) \propto p(hu_m|a', b') \prod_u p(\xi_u|a, hu_m) \propto hu_m^{a'+\aleph_m a-1} exp\{-(b' + \sum_u \xi_u)hu_m\} \tag{6}$$

$$p(hi_n|c', d', \eta_i) \propto p(hi_n|c', d') \prod_i p(\eta_i|c, hi_n) \propto hi_n^{c'+\chi_n c-1} exp\{-(d' + \sum_i \eta_i)hi_n\} \tag{7}$$

where $\aleph_m$ is the number of users having attribute $m$ and $\chi_n$ is the number of items that have attribute $n$. Thus, the posterior Gamma distribution of $hu_m$ and $hi_n$ are

$$hu_m|\xi_u \sim Gamma(a' + \aleph_m a, b' + \sum_u \xi_u) \tag{8}$$

$$hi_n|\eta_i \sim Gamma(c' + \chi_n c, d' + \sum_i \eta_i) \tag{9}$$

$hu_m$ is affected by $\aleph_m$ and the latent user preference (i.e., $\xi_u$) and $hi_n$ is affected by $\chi_n$ and the latent item attractiveness (i.e., $\eta_i$)

We then update the variational *shape* ($shp$) and *rate* ($rte$) parameters for the user/item metadata.

$$(\zeta_m^{shp}, \zeta_m^{rte}) = (a' + \aleph_m a, b' + \sum_u \frac{\kappa_u^{shp}}{\kappa_u^{rte}}) \tag{10}$$

$$(\rho_n^{shp}, \rho_n^{rte}) = (c' + \chi_n c, d' + \sum_i \frac{\tau_i^{shp}}{\tau_i^{rte}}) \tag{11}$$

### 4.2 Derivation of User Latent Preference and Item Latent Attractiveness

With the Gamma distribution of latent user preference and latent item attractiveness:

$$p(\xi_u|a, hu_m) \propto \xi_u^{a-1} exp\{-(\prod_{m=1}^{M} hu_m^{fu_{u,m}})\xi_u\} \tag{12}$$

$$p(\eta_i|c, hi_n) \propto \eta_i^{c-1} exp\{-(\prod_{n=1}^{N} hi_n^{fi_{i,n}})\eta_i\} \tag{13}$$

With the Gamma distribution of user/item's global static factor:

$$p(\overline{\theta}_{uk}|b, \xi_u) \propto \xi_u^b exp\{-\xi_u\overline{\theta}_{uk}\} \tag{14}$$

$$p(\overline{\beta}_{ik}|d, \eta_i) \propto \eta_i^d exp\{-\eta_i\overline{\beta}_{ik}\} \tag{15}$$

The posterior probability of $\xi_u$ and $\eta_i$ becomes:

$$p(\xi_u|a, hu_m, \overline{\theta}_{uk}) \propto p(\xi_u|a, hu_m) \prod_k p(\overline{\theta}_{uk}|b, \xi_u) \propto \xi_u^{a+Kb-1} exp\{-(\prod_{m=1}^{M} hu_m^{fu_{u,m}} + \sum_k \overline{\theta}_{uk})\xi_u\} \tag{16}$$

$$p(\eta_i|c, hi_n, \overline{\beta}_{ik}) \propto p(\eta_i|c, hi_n) \prod_k p(\overline{\beta}_{ik}|c, \eta_i) \propto \eta_i^{c+Kd-1} exp\{-(\prod_{n=1}^{N} hi_n^{fi_{i,n}} + \sum_k \overline{\beta}_{ik})\eta_i\} \quad (17)$$

where $K$ is the number of latent components. Thus, the posterior Gamma distribution of $\xi_u$ and $\eta_i$ are

$$\xi_u|\overline{\theta}_{uk} \sim Gamma(a + Kb, \prod_{m=1}^{M} hu_m^{fu_{u,m}} + \sum_k \overline{\theta}_{uk}) \quad (18)$$

$$\eta_i|\overline{\beta}_{ik} \sim Gamma(c + Kd, \prod_{n=1}^{N} hi_n^{fi_{i,n}} + \sum_k \overline{\beta}_{ik}) \quad (19)$$

We then update the variational *shape* ($shp$) and *rate* ($rte$) parameters for the latent user preference and latent attractiveness:

$$(\kappa_u^{shp}, \kappa_u^{rte}) = (a + Kb, \prod_{m=1}^{M} (\frac{\zeta_m^{shp}}{\zeta_m^{rte}})^{fu_{u,m}} + \sum_k \frac{\overline{\nu}_{uk}^{shp}}{\overline{\nu}_{uk}^{rte}}) \quad (20)$$

$$(\tau_i^{shp}, \tau_i^{rte}) = (c + Kd, \prod_{n=1}^{N} (\frac{\rho_n^{shp}}{\rho_n^{rte}})^{fi_{i,n}} + \sum_k \frac{\overline{\mu}_{ik}^{shp}}{\overline{\mu}_{ik}^{rte}}) \quad (21)$$

### 4.3 Derivation of the Rating

To ensure the conjugacy of the model structure, inspired by [2, 1, 4, 3], the rating $y_{ui,t}$ is replaced with auxiliary latent variable $z_{ui,k,t} \sim Poisson((\theta_{uk,t} + \overline{\theta}_{uk})(\beta_{ik,t} + \overline{\beta}_{ik}))$. With the additive property of Poisson distribution, $y_{ui,t}$ is expressed as $y_{ui,t} = \sum_k z_{ui,k,t}$. The complete conditional for the auxiliary latent variable vector is

$$z_{ui,k,t}|\theta_{uk,t}, \overline{\theta}_{uk}, \beta_{ik,t}, \overline{\beta}_{ik} \sim Mult\left(y_{ui,t}, \frac{(\theta_{uk,t} + \overline{\theta}_{uk})(\beta_{ik,t} + \overline{\beta}_{ik})}{\sum_k (\theta_{uk,t} + \overline{\theta}_{uk})(\beta_{ik,t} + \overline{\beta}_{ik})}\right) \quad (22)$$

Thus, the update for the variational multinominal is

$$\phi_{ui,t,k} = (exp\{\Psi(\nu_{uk,t}^{shp}) - log(\nu_{uk,t}^{rte})\} + exp\{\Psi(\overline{\nu}_{uk}^{shp}) - log(\overline{\nu}_{uk}^{rte})\})$$
$$* (exp\{\Psi(\mu_{ik,t}^{shp}) - log(\mu_{ik,t}^{rte})\} + exp\{\Psi(\overline{\mu}_{ik}^{shp}) - log(\overline{\mu}_{ik}^{rte}))\}) \quad (23)$$

where $\Psi(.)$ is the *digamma* function. This update comes from the expectation of the log of a Gamma variable, for example $E_q[log(\theta_{uk,t})] = \Psi(\nu_{uk,t}^{shp}) - log(\nu_{uk,t}^{rte})$.

### 4.4 Derivation of the Global Static Factor of User/Item

The global static factor of user/item is affected by the latent user preference/latent item attractiveness, the local dynamic factors and the observations with the conjugate Gamma-Poisson distribution. Thus, the complete conditional of those vectors are

$$\overline{\theta}_{uk}|\overline{\beta}_{ik}, \beta_{ik,t}, \xi_u, z_{ui,k,t} \sim Gamma(b + \sum_{i,t} z_{ui,k,t}, \xi_u + \sum_i (\overline{\beta}_{ik} + \sum_t \beta_{ik,t})) \quad (24)$$

$$\overline{\beta}_{ik}|\overline{\theta}_{uk}, \theta_{uk,t}, \eta_i, z_{ui,k,t} \sim Gamma(c + \sum_{u,t} z_{ui,k,t}, \eta_i + \sum_u (\overline{\theta}_{uk} + \sum_t \theta_{uk,t})) \quad (25)$$

The update for variational global static factor of user/item becomes

$$(\overline{\nu}_{uk}^{shp}, \overline{\nu}_{uk}^{rte}) = (b + \sum_{i,t} y_{ui,t}\phi_{ui,t,k}, \frac{\kappa_u^{shp}}{\kappa_u^{rte}} + \sum_i (\frac{\overline{\mu}_{ik}^{shp}}{\overline{\mu}_{ik}^{rte}} + \sum_t \frac{\mu_{ik,t}^{shp}}{\mu_{ik,t}^{rte}})) \quad (26)$$

$$(\overline{\mu}_{ik}^{shp}, \overline{\mu}_{ik}^{rte}) = (d + \sum_{u,t} y_{ui,t}\phi_{ui,t,k}, \frac{\tau_i^{shp}}{\tau_i^{rte}} + \sum_u (\frac{\overline{\nu}_{uk}^{shp}}{\overline{\nu}_{uk}^{rte}} + \sum_t \frac{\nu_{uk,t}^{shp}}{\nu_{uk,t}^{rte}})) \quad (27)$$

## 4.5 Derivation of the Local Dynamic Factor of User/Item

**Initial states of the local dynamic factor of user/item** are affected by the observations, the next auxiliary variables and the global static factors with the conjugate Gamma-Poisson distribution. Thus, the complete conditionals of those vectors are

$$\theta_{uk,1}|\overline{\beta}_{ik}, \beta_{ik,1}, \lambda_{uk,1}, z_{ui,k,1} \sim Gamma(a_\theta + a_\lambda + \sum_i z_{ui,k,1}, a_\theta b_\theta + a_\lambda \lambda_{uk,1} + \sum_i (\overline{\beta}_{ik} + \beta_{ik,1}))$$

(28)

$$\beta_{ik,1}|\overline{\theta}_{uk}, \theta_{uk,1}, \iota_{ik,1}, z_{ui,k,1} \sim Gamma(a_\beta + a_\iota + \sum_u z_{ui,k,1}, a_\beta b_\beta + a_\iota \iota_{ik,1} + \sum_u (\overline{\theta}_{uk} + \beta_{uk,1}))$$

(29)

The update of variational of initial states of local dynamic factor of user/item becomes:

$$(\nu_{uk,1}^{shp}, \nu_{uk,1}^{rte}) = (a_\theta + a_\lambda + \sum_i y_{ui,1}\phi_{ui,1,k}, a_\theta b_\theta + a_\lambda \frac{\gamma_{uk,1}^{shp}}{\gamma_{uk,1}^{rte}} + \sum_i (\frac{\overline{\mu}_{ik}^{shp}}{\overline{\mu}_{ik}^{rte}} + \frac{\mu_{ik,1}^{shp}}{\mu_{ik,1}^{rte}}))$$

(30)

$$(\mu_{ik,1}^{shp}, \mu_{ik,1}^{rte}) = (a_\beta + a_\iota + \sum_u y_{ui,1}\phi_{ui,1,k}, a_\beta b_\beta + a_\iota \frac{\omega_{ik,1}^{shp}}{\omega_{ik,1}^{rte}} + \sum_u (\frac{\overline{\nu}_{uk}^{shp}}{\overline{\nu}_{uk}^{rte}} + \frac{\nu_{uk,1}^{shp}}{\nu_{uk,1}^{rte}}))$$

(31)

**Auxiliary variables of the local dynamic factor of user/item** are affected by the previous local dynamic factor, the next local dynamic factor with the conjugate Gamma-Gamma distribution. Thus, the complete conditionals of those vectors are

$$\lambda_{uk,t}|\theta_{uk,t}, \theta_{uk,t+1} \sim Gamma(a_\lambda + a_\theta, a_\lambda \theta_{uk,t} + a_\theta \theta_{uk,t+1})$$

(32)

$$\iota_{ik,t}|\beta_{ik,t}, \beta_{ik,t+1} \sim Gamma(a_\iota + a_\beta, a_\iota \beta_{ik,t} + a_\beta \beta_{ik,t+1})$$

(33)

The update for variational of auxiliary variables of local dynamic factor of user/item becomes

$$(\gamma_{uk,t}^{shp}, \gamma_{uk,t}^{rte}) = (a_\lambda + a_\theta, a_\lambda \frac{\nu_{uk,t}^{shp}}{\nu_{uk,t}^{rte}} + a_\theta \frac{\nu_{uk,t+1}^{shp}}{\nu_{uk,t+1}^{rte}})$$

(34)

$$(\omega_{ik,t}^{shp}, \omega_{ik,t}^{rte}) = (a_\iota + a_\beta, a_\iota \frac{\mu_{ik,t}^{shp}}{\mu_{ik,t}^{rte}} + a_\beta \frac{\mu_{ik,t+1}^{shp}}{\mu_{ik,t+1}^{rte}})$$

(35)

**Local dynamic factor of user/item at time** $t > 1$ are affected by the observations, the previous auxiliary variables, the next auxiliary variables and the global static factors with the conjugate Gamma-Poisson distribution. Thus, the complete conditionals of those vector are

$$\theta_{uk,t}|\overline{\beta}_{ik}, \beta_{ik,t}, \lambda_{uk,1}, z_{ui,k,t} \sim Gamma(a_\theta + a_\lambda + \sum_i z_{ui,k,t}, a_\theta \lambda_{uk,t-1} + a_\lambda \lambda_{uk,t} + \sum_i (\overline{\beta}_{ik} + \beta_{ik,t}))$$

(36)

$$\beta_{ik,t}|\overline{\theta}_{uk}, \theta_{uk,t}, \iota_{ik,1}, z_{ui,k,t} \sim Gamma(a_\beta + a_\iota + \sum_u z_{ui,k,t}, a_\beta \iota_{ik,t-1} + a_\iota \iota_{ik,t} + \sum_u (\overline{\theta}_{uk} + \beta_{uk,t}))$$

(37)

The update for variational of local dynamic factor of user/item at time $t > 1$ becomes

$$(\nu_{uk,t}^{shp}, \nu_{uk,t}^{rte}) = (a_\theta + a_\lambda + \sum_i y_{ui,t}\phi_{ui,t,k}, a_\theta \frac{\gamma_{uk,t-1}^{shp}}{\gamma_{uk,t-1}^{rte}} + a_\lambda \frac{\gamma_{uk,t}^{shp}}{\gamma_{uk,t}^{rte}} + \sum_i (\frac{\overline{\mu}_{ik}^{shp}}{\overline{\mu}_{ik}^{rte}} + \frac{\mu_{ik,t}^{shp}}{\mu_{ik,t}^{rte}}))$$

(38)

$$(\mu_{ik,t}^{shp}, \mu_{ik,t}^{rte}) = (a_\beta + a_\iota + \sum_u y_{ui,t}\phi_{ui,t,k}, a_\beta \frac{\omega_{ik,t-1}^{shp}}{\omega_{ik,t-1}^{rte}} + a_\iota \frac{\omega_{ik,t}^{shp}}{\omega_{ik,t}^{rte}} + \sum_u (\frac{\overline{\nu}_{uk}^{shp}}{\overline{\nu}_{uk}^{rte}} + \frac{\nu_{uk,t}^{shp}}{\nu_{uk,t}^{rte}}))$$

(39)