[Reviews · NeurIPS 2018]

Reviewer 1



Summary: The authors develop a Gamma-Poisson factorization model that includes metadata and models user preferences and item attractiveness in a dynamic context. They develop a variational inference algorithm and demonstrate that their approach outperforms other methods on five data sets. Quality: The technical quality of this work appears to be sound. For evaluation, the metrics used are in line with the way these systems are actually deployed (e.g., rank-based instead of just RMSE of the ratings). [Thank you!!] I do have three notes, however: 1. I think the authors sell Gaussian MF a little short. While I really like Gamma-Poisson models, the MF-only part of the collaborative topic regression model (http://www.cs.columbia.edu/~blei/papers/WangBlei2011.pdf, has source code available) outperforms PF (using the code associated with [19]) on similar held-out prediction tasks shown in the submitted paper. This indicated that the PF > Gaussian MF isn’t as crisp as indicated in [19] or the submitted paper. 2. The conjugate relationships of the Gamma variables (using one gamma variable as the rate of another gamma variable) appear to be primarily for convenience in inference. There is nothing inherently wrong with this other than the terms used to describe the variables won’t always match with a reader’s naive intuition. For example, we draw the weight of the metadata for users hu_m and this is used as the shape parameter to draw user preferences. A reader might naively think that a higher weight hu_m would lead to higher user preferences, but the inverse is true since we are using the shape/rate parameterization. There’s also a complicated relationship with the variance. It might be worth justifying these choices a bit better and perhaps being more explicit about nomenclature and the relationships between variables for the sake of the reader. 3. There are more example of incorporating different kinds of metadata in PF, e.g., - Do, Trong Dinh Thac, and Longbing Cao. "Coupled Poisson Factorization Integrated with User/Item Metadata for Modeling Popular and Sparse Ratings in Scalable Recommendation." AAAI2018 (2018): 1-7. - Chaney, Allison JB, David M. Blei, and Tina Eliassi-Rad. "A probabilistic model for using social networks in personalized item recommendation." In Proceedings of the 9th ACM Conference on Recommender Systems, pp. 43-50. ACM, 2015. Clarity: The submission is dense, but generally clear. Fonts on the figures should be larger. Originality: The proposed model is new, but the authors’ general approach is fairly intuitive and in line with previous work. Additionally, since the model is conjugate, the inference algorithm is relatively straightforward to derive. Still, for such a complicated model, there is merit in formalizing the model, executing the inference derivations, and validating the model. Significance: The submission addresses a difficult task and provides evidence for the superiority of their approach. The application domain is narrow in some sense, but still broad enough for this work to have real-world impact by being incorporated into live recommendation systems. The significance of this work would be greater if source code was released alongside the paper. Confidence score: I rarely like to mark that I am “absolutely certain” about a paper, but I have expertise in both the application domain as well as the modeling and inference approaches.

Reviewer 2



This paper presents a dynamic Poisson factorisation model for recommendation system, which incorporates meta information of users and items. Specifically, meta information of users and items is incorporated in the gamma scale parameter of the static portions of users and items, respectively; the dynamics are modelled by a Markov chain where the dynamic portions are conditioned on the portions of the previous step. The inference of the model is done by stochastic variational inference. The experiments show that the model has better prediction performance than several previous models on dynamical data. Quality: In general, this paper is in good quality. (1) The motivations for using meta information in recommendation systems are quite intuitive. (2) The related work and background have been properly covered. (3) The experiments are relatively strong. The datasets are well selected, the results look promising, and the analysis is comprehensive. To me, this is where the credits come from. (4) The main issue of the paper is the novelty of the model and the inference. The model structure of using gamma parameters to incorporate meta information has been explored in several previous works and this paper didn't provide a structure with much differences to the previous ones. The same thing happens to the model structure of modelling dynamical data, which is similar to the one in DCPF. The contribution of this paper is combining the two structures into one model with a summation. However, the idea of summation has also been proposed, as in “Content-based recommendations with Poisson factorization”. Using SVI for inference can hardly be counted as the contribution of this paper. (5) One minor issue is that, to me, the proposed GDMF (the one without meta info) is similar to DCPF, in terms of model structure. But GDMF is reported to have better performance than DCPF. Can the authors clearly point out the differences between GDMF and DCPF, as well as analyse where the performance gain come from? Clarity: The paper is easy to follow. Originality and significance: As I've discussed, the originality is the main issue. The novelty of the model structure and the inference is limited. But the experiments and results look strong. Therefore, to me, the main significance of the paper is on the empirical results. -------------------------------------------------------------------------------------------- Thanks for the authors' response, which addresses my concern about the differences between GDMF and DCPF. I hope the this explanation can be added in the revision of the paper.

Reviewer 3



Paper : 2816 This paper presents a scalable model which integrates dynamical Poisson factorization with metadata information. Here metadata information contains non-dynamic users and items attributes. Authors design an algorithm which has two stages. One is to incorporate the metadata information to static latent random variables for each user and item. Second part is to learn dynamical nature of the observations via latent single order markov chain variable of gamma-gamma structure. The markov chain is modeled by absorbing the dynamical nature of data via its rate parameter of gamma distribution. Stochastic Variational inference has been used to infer approximating distribution’s parameters. Algorithm has shown having learn-ability and predictive power over large datasets, also it has shown how it would enhance the recommendation of more sparsely rated items to users compared to baseline algorithms. Main contribution of paper is to model how to incorporate effect of users and items attributes which have nature of categorical distribution as an weight which is sampled from gamma distribution. Experiments has been done on three different datasets, results show advantage of current algorithm than baselines, there are some discussion about results and clarity which will be detailed in following. Quality: Paper has clear approach to present the model and dependencies between latent variables and observation data. Most of derivation have been verified. Just there are few minor points: -in line 103 and 107, it has not been discussed how to determine the value for f_{u,m} or f_{i,n}, it is guessed to be that f_{u,m} and f_{i,n} are binary values that indicates presence or lack of presence of an attribute for an item or user in certain datasets, but has not been specified. -in results section, in order to verify effect of metadata incorporation, it is useful to plot the GDMF recommendation beside mGDMF and DCPF in Figure 2. This way we can see how much recommending sparsely rated item is resulted by incorporation of metadata attributes to the model. Clarity: Paper has been motivated well and general concepts and background have been explained well, just a few minor points: - Most of “deviation”s in supplementary material section should be changed to “derivation” - on line 110 to 114, it is discussed about necessity of introducing auxiliary variable, and reason mentioned to be not attaining positive correlation between latent variables in Markov chain. It will be useful for reader if it is explained why such thing happens and how auxiliary variable can solve this problem. - adding graphical diagram (plate model) to the paper makes the article an easier read Originality: As mentioned in summary the main contribution of this paper is to model incorporation of users and items attributes which have categorical nature as an importance weight which is sampled from gamma distribution and it will affect the item or user latent variable. Authors have cited many of recent works, and have done an in depth literature review also could distinguish their contribution to what has been done so far clearly. Significance: As discussed, in Quality section, it is useful in figure 2 to have another subplot for GDMF recommendation to verify the significance of mGDMF. Also I would like to mention that dynamical Poisson factorization model in [31] is close to this dynamical Poisson factorization system and code is available on github that could have been used as one of baseline methods.